# Silica-Polymer Composites as the Novel Antibiotic Delivery Systems for Bone Tissue Infection

**DOI:** 10.3390/pharmaceutics12010028

**Published:** 2019-12-30

**Authors:** Adrianna Skwira, Adrian Szewczyk, Agnieszka Konopacka, Monika Górska, Dorota Majda, Rafał Sądej, Magdalena Prokopowicz

**Affiliations:** 1Department of Physical Chemistry, Faculty of Pharmacy, Medical University of Gdańsk, Hallera 107, 80-416 Gdańsk, Poland; adrianna.skwira@gumed.edu.pl (A.S.); adrian.szewczyk@gumed.edu.pl (A.S.); 2Department of Pharmaceutical Microbiology, Faculty of Pharmacy, Medical University of Gdańsk, Hallera 107, 80-416 Gdańsk, Poland; agnieszka.konopacka@gumed.edu.pl; 3Department of Molecular Enzymology and Oncology, Intercollegiate Faculty of Biotechnology, University of Gdańsk and Medical University of Gdańsk, 80-210 Gdańsk, Poland; monika.gorska@gumed.edu.pl (M.G.); rsadej@gumed.edu.pl (R.S.); 4Faculty of Chemistry, Jagiellonian University, Gronostajowa 2, 30-387 Kraków, Poland; majda@chemia.uj.edu.pl

**Keywords:** drug delivery system, mesoporous silica, silica-polymer, ciprofloxacin, polydimethylsiloxane, composites, coating blend

## Abstract

Bone tissue inflammation, *osteomyelitis*, is commonly caused by bacterial invasion and requires prolonged antibiotic therapy for weeks or months. Thus, the aim of this study was to develop novel silica-polymer local bone antibiotic delivery systems characterized by a sustained release of ciprofloxacin (CIP) which remain active against *Staphylococcus aureus* for a few weeks, and do not have a toxic effect towards human osteoblasts. Four formulations composed of ethylcellulose (EC), polydimethylsiloxane (PDMS), freeze-dried CIP, and CIP-adsorbed mesoporous silica materials (MCM-41-CIP) were prepared via solvent-evaporation blending method. All obtained composites were characterized in terms of molecular structure, morphological, and structural properties by using Fourier Transform Infrared Spectroscopy (FTIR), scanning electron microscopy equipped with energy-dispersive X-ray spectroscopy (SEM/EDX), and X-ray diffraction (XRD), thermal stability by thermogravimetric analysis (TGA) and differential scanning calorimetry (DSC), and in vitro antibiotic release. The antibacterial activity against *Staphylococcus aureus* (ATCC 6538) as well as the in vitro cytocompatibility to human osteoblasts of obtained composites were also examined. Physicochemical results confirmed the presence of particular components (FTIR), formation of continuous polymer phase onto the surface of freeze-dried CIP or MCM-41-CIP (SEM/EDX), and semi-crystalline (composites containing freeze-dried CIP) or amorphous (composites containing MCM-41-CIP) structure (XRD). TGA and DSC analysis indicated the high thermal stability of CIP adsorbed onto the MCM-41, and higher after MCM-41-CIP coating with polymer blend. The release study revealed the significant reduction in initial burst of CIP for the composites which contained MCM-41-CIP instead of freeze-dried CIP. These composites were also characterized by the 30-day activity against *S. aureus* and the highest cytocompatibility to human osteoblasts in vitro.

## 1. Introduction

Surgical site infection, bone fracture, or trauma may lead to severe bone inflammation such as *osteomyelitis* [1]. Treatment with systemic delivery of antibiotics usually lasts from 4 to 6 weeks, and requires an administration of high dosages to achieve a sufficient concentration at the site of infection. Moreover, it may be unachievable in patients with poorly vascularized infected tissue and osteonecrosis, which are common symptoms accompanying *osteomyelitis* [2].

Therefore, local bone antibiotic delivery systems (LBADS) have gained an increasing interest in the treatment of bone tissue infections, as an alternative method to the systemic therapy [3,4], that provide the release of antibiotics in controlled and sustained manner directly in infected bone site. Therapy with the use of LBADS has been found as safe and effective due to limiting its action only to desired site. Local concentrations of antibiotics that have been released from implantable LBADS are many times higher than minimal inhibitory concentration (MIC). Thus, LBADS provide the efficient antibacterial activity and decrease the risk of bacterial resistance at the site of infection. However, the maintenance of antibiotics at high concentrations in bone tissue for a long time may lead to significant impairment of bone cells functions or cytotoxic effect [5,6]. Duewelhenke et al. [7] and Rathbone et al. [8] examined the influence of antimicrobial agents on human osteoblasts activity and viability. The authors presented a concentration-dependent cytotoxicity of antibiotics towards human osteoblasts. Therefore, the greatest challenge for LBADS is to provide the antibiotic at effective, bactericidal levels which are not toxic to human tissues. 

Among LBADS, mesoporous silica materials (MSM) have received considerable attention, due to their unique properties, such as high surface area, controlled mesopore size, tunable pore diameter, thermal stability, biocompatibility, and modified release profile of active substance [9,10,11]. However, the release profiles of water-soluble drug, such as antibiotics, loaded into MSM have usually presented high-level burst release [12] which may be difficult to control and lead to cytotoxic effect. Therefore, the development of antibiotic-loaded mesoporous silica characterized by a sustained release with reduced initial burst has constituted a major challenge. Various methods of chemical drug release modification have been described [13,14,15,16]. They have been mainly focused on mesoporous silica surface functionalization by the reaction of silanols with the amino-propyl [17], carboxylic [18] groups, or organic chains [19,20]. These processes may slow down the release rate but usually the time of drug release does not exceed a few days [21,22]. However, due to the long duration of *osteomyelitis* treatment, there is a need to provide the local system which releases the antibacterial agents for at least a few weeks. Presently, a great attention has been paid to fabrication of silica-based composites [23,24] which exhibit both the significantly prolonged release of drug and good biocompatibility [25,26]. Advanced silica-polymer composites have been mainly prepared by the blending method combined with the solvent evaporation [27]. 

Therefore, the main objective of this study was to design the silica-polymer composites characterized by (i) reduced initial burst, (ii) sustained release of antibiotic, which remain active for a few weeks against bacteria that commonly cause *osteomyelitis*, and (iii) do not have a toxic effect towards human osteoblasts. *Staphylococcus aureus* strains have been the most frequently isolated pathogens from the site of infection in bone tissue [28]. These strains are sensitive to fluoroquinolones which present essential characteristics for the use in local drug delivery systems e.g., favorable penetration into the bone tissue and stability at body temperature [29,30]. Among fluoroquinolones, ciprofloxacin presents one of the highest activity against strains of *Staphylococcus aureus* and *Pseudomonas aeruginosa* [31]. Therefore, ciprofloxacin was chosen as an active component loaded onto MSM, type of MCM-41. The coating blend was composed of ethylcellulose (EC) and polydimethylsiloxane (PDMS). EC as the most stable, non-toxic cellulose derivative with modifiable viscosity has been broadly used in pharmaceuticals as a good film forming agent [32]. Moreover, one of the widely known applications of EC is film coating in dosage forms with controlled drug release [33,34]. PDMS was chosen due to its biocompatibility, bioinertness, and documented applicability in implantable materials [35,36]. Additionally, unique physicochemical properties such as high elasticity (flexibility), adhesion and hydrophobicity play a crucial role in considering PDMS as valuable excipient for drug release modification from dosage forms [37,38]. Nahrup et al. [39] indicated that the usage of PDMS as a pharmaceutical tablet coating may provide a possible zero-order release (highly desirable delivery of a constant amount of drug per unit of time). 

In this paper, synthesized MSM, type MCM-41, adsorbed with ciprofloxacin (MCM-41-CIP) were coated either with EC or blend of EC and PDMS. All the composites were evaluated in terms of physicochemical properties, release study in vitro, antimicrobial activity (against *Staphylococcus aureus*), and cytocompatibility in vitro (osteoblastic cell line) to assess their potential applicability in vivo.

## 2. Materials and Methods

### 2.1. Materials

Tetraethyl orthosilicate (TEOS), cetyltrimethylammonium bromide (CTAB), ethanol, aqueous ammonia (25%), hydroxyl-terminated polydimethylsiloxane (PDMS, 150 cSt, d = 0.97 mg/mL) were purchased from Sigma-Aldrich (Saint Louis, MO, USA). Ethylcellulose (EC, Ethocel 20 cP, molecular weight: 454 g/mol, ethoxyl content: 48.0%–49.5% wt.) was obtained from Dow Chemical (Midland, MI, USA). Acidic solution of ciprofloxacin (pH = 3.5, 10 mg/mL, Proxacin) was obtained from Polfa S.A. (Warsaw, Poland). Mueller Hinton Broth, 1:1 mixture of Ham’s F12 Medium Dulbecco’s Modified Eagle’s Medium, 10% fetal bovine serum, and penicillin/streptomycin (10,000 U/mL/10 mg/mL), DNA staining with 4,6-diamidino-2-phenylindoledihydrochloride (DAPI) were purchased from Sigma-Aldrich. MycoAlertTM Mycoplasma Detection Kit was obtained from Lonza (Basel, Switzerland). Human fetal osteoblastic cell line (hFOB 1.19) was obtained from American Type Culture Collection (cat. no. CRL-11372), pLVTHM was a gift from Didier Trono (Addgene plasmid # 12247).

### 2.2. Synthesis of Mesoporous Silica Materials (MCM-41)

The synthesis of MCM-41 was performed using sol–gel method as previously described [40] with the usage of TEOS as a silica source and cetyltrimethylammonium bromide (CTAB) cationic surfactant as a structure directing agent [36]. The water, ethanol, aqueous ammonia (25%), and CTAB in the amounts of 125, 12.5, 9.18 and 2.39 g, respectively, were mixed together in polypropylene beaker by stirring for 10 min (300 rpm, 25 °C) until complete dissolution of surfactant. Then, TEOS in amount of 10.03 g was added, and the resulting mixture was continuously stirred for 2 h. Next, the mixture was aging at 90 °C for 5 days without stirring. Once the solid product occurred, it was washed with absolute ethanol and dried at 40 °C for 1 h. To remove CTAB, the calcination process was performed (550 °C, 6 h, heating rate of 1 °C/min) in a muffle furnace (FCF 7SM, CZYLOK, Jastrzebie-Zdroj, Poland). The final MSM MCM-41 were micronized (50 rpm, 10 min, Mortar Grinder Pulverisette 2, Fritsch, Weimar, Germany) to obtain the fraction size ranging from 200–500 µm for further experiments.

### 2.3. Ciprofloxacin Adsorption

The ciprofloxacin (CIP) adsorption onto the mesoporous silica MCM-41 was carried out by using previously optimized procedure [40]. Briefly, MCM-41 material (particles fraction in the range of 200–500 µm) was suspended in the 10 mg/mL CIP lactate solution (pH = 3.5) using 50:1 mass to volume ratio. Then, CIP-loaded MCM-41 material (MCM-41-CIP) was centrifuged, separated from supernatant, and freeze-dried (−52 °C, 0.1 mBar, 24 h). The concentration of the CIP remaining in the supernatant was calculated spectrophotometrically at 278 nm (model UV-1800 UV–Vis spectrophotometer, Shimadzu, Kyoto, Japan), whereas the amount of CIP adsorbed onto MCM-41 at equilibrium state and adsorption efficiency were calculated using Equations (1) and (2), respectively:(1)$$Q_{e}=\frac{(C_{0}-C_{e})\cdot V}{m},$$
(2)$$\%Ads=\frac{C_{0}-C_{e}}{C_{e}}\;\times \;100\%,\;$$
*Q_e_* [mg/g]—an amount of CIP adsorbed onto MCM-41 at the equilibrium state; %*Ads* [%]—an adsorption efficiency coefficient; *C*_0_ [mg/mL]—an initial CIP concentration; *C_e_* [mg/mL]—a CIP concentration at equilibrium state; *V* [mL]—a volume of CIP solution, *m* [g]—a mass of MCM-41. 

The adsorption process was repeated 6 times and both the amount of CIP adsorbed and the adsorption efficiency were expressed as the mean values ± SD. The charge change of MCM-41 surface after adsorption process of CIP was performed by using the zeta potential analysis (Litesizer 500, Anton-Paar, Graz, Austria). To measure the zeta potential the samples of both the MCM-41 (before CIP adsorption) and the MCM-41-CIP (after CIP adsorption) were immersed in HCl solution (pH = 3.5) at concentration of 1 mg/mL and dispersed in ultrasonic bath for 5 min. The zeta potential values were expressed as a mean ± SD calculated from 10 measurements. The parameters characterizing the porosity of the MCM-41 before and after CIP adsorption were determined by the measurements of low-temperature nitrogen adsorption–desorption at −196 °C using a volumetric adsorption analyzer ASAP 2405 (Micromeritics, Norcros, GA, USA). Prior to the experiment, all the samples were dried overnight at 25 °C under vacuum. The specific surface areas were calculated using the standard Brunauer–Emmett–Teller (BET) equation for nitrogen adsorption data acquired in the range of relative pressure p/p_0_ in the range from 0.05 to 0.25. The total pore volumes were estimated from a single point adsorption at 0.993 p/p_0_. The average pore size was determined from the desorption branch of the nitrogen isotherm using the Barrett–Joyner–Halenda (BJH) procedure.

### 2.4. Composites Fabrication

Consecutive stages of composites fabrication were presented in Scheme 1. The composites named respectively EC/CIP, EC/PDMS/CIP, EC/MCM-41-CIP, and EC/PDMS/MCM-41-CIP were prepared via solvent-evaporation blending method. The qualitative and quantitative composition data of each composite were presented in Table 1. In brief, to prepare EC/PDMS/CIP composites (Scheme 1a), PDMS in the volume of 5 µL was mixed with 245 µL of EC ethanolic solution (5% (*w*/*w*)). Then, 0.79 ± 0.05 mg of freeze-dried CIP in quantity corresponding to the amount of CIP adsorbed onto 6 ± 0.02 mg of MCM-41-CIP was added into the sol of EC/PDMS at dynamic viscosity of ~49 ± 4 mPa·s, 24 °C (Rotational Viscometer V2-L, Conbest, Krakow, Poland). Composites containing MCM-41-CIP (EC/PDMS/MCM-41-CIP) were fabricated in the same manner but instead of freeze-dried CIP, 6 mg ± 0.02 mg of MCM-41-CIP (corresponding to 0.79 mg of adsorbed drug) was added into the EC/PDMS sol (Scheme 1b). In case of composites without PDMS (EC/CIP, EC/MCM-41-CIP), freeze-dried CIP or MCM-41-CIP were added directly to 250 µL of EC ethanolic solution (5% (*w*/*w*)) at dynamic viscosity of ~44 ± 3 mPa·s, 24 °C. All obtained suspensions were sonicated in ice bath for 20 min, poured into the polypropylene molds, and incubated at 30 ± 0.5 °C till complete ethanol evaporation (24 h). Composites were then removed from the molds, weighted (Quintix 1250, Sartorius Lab, Goettingen, Germany), and stored in the desiccator (25 °C). The amount of CIP was 0.79 ± 0.05 mg per each composite.

### 2.5. Composites Physicochemical Characterization

All obtained composites were investigated in terms of molecular structure with the use of Fourier Transform Infrared Spectroscopy (FTIR, Jasco FT/IR-4200, Jasco, Pfungstadt, Germany) in the range of 4000–400 cm^−1^. Surface morphology and elemental analysis of composites were investigated using a scanning electron microscope with energy-dispersive X-ray spectroscopy (SEM/EDX, Hitachi SU-70, Tokyo, Japan). The results were obtained by using electron microscope at an acceleration voltage of 3 kV. The crystallinity of composites was characterized by X-ray diffraction analysis (XRD, Empyrean PANalytical, Malvern, UK). The diffractometer was operated using Cu Kα radiation beam at 40 kV and 40 mA, in the 2*θ* range between 5 and 40° with the following parameters: a step width of 0.020° and a scanning rate of 0.5°/min. The texture properties were determined by TA. XTplusC Texture Analyzer (Godalming, UK). TGA characterization was performed using a Mettler Toledo TGA/SDTA 851e apparatus (Warsaw, Poland), calibrated with indium, zinc, and aluminum (accuracy equal to 10^−6^ g). The samples were placed in alumina crucible and heated from 25 to 1000 °C with rate of 10 K min^−1^, in argon atmosphere (60 cm^3^ min^−1^). The DSC measurements were performed using Mettler Toledo apparatus DSC 821e (Warsaw, Poland) equipped with the IntraCooler system. Calibration for the heat flux and temperature was done with indium and zinc standards. The sample was placed in aluminum pan and heated from −40 to 300 °C in the Ar (60 cm^3^ min^−1^).

### 2.6. Ciprofloxacin In Vitro Release

For the drug release study, each composite containing 0.79 ± 0.05 mg of CIP was immersed in 2 mL of distilled water (pH = 7.0), and continuingly shaken at 37 ± 0.5 °C (80 rpm). The theoretical highest concentration of drug after complete CIP release into the medium was below the 10% drug aqueous solubility (solubility of ciprofloxacin lactate around 100 mg/mL [41]) providing sink conditions. The amounts of CIP released were measured every 24 h for 30 days. The whole release medium was replaced after each measurement to simulate dynamic fluid conditions in the body. Quantitative determinations of the amount of CIP released was based on pre-calibration of the spectrometer at 278 nm wavelength using standard solutions of the CIP. The release study was repeated 6 times and values were given as mean ± SD.

### 2.7. Composites Biological Evaluation

#### 2.7.1. Composites Sterilization

The composites intended for the biological evaluation were prepared via aseptic assembly [42], which is a pharmacopeial method dedicated for the materials that cannot be terminally sterilized, such as antibiotic-loaded systems. MCM-41 was sterilized by the heating in air for a period of 3 h at 300 °C in a muffle furnace (FCF 7SM, CZYLOK, Jastrzebie-Zdroj, Poland). The CIP solution used to adsorption process was sterilized by producer. Additionally, the sterility of the composites was provided by the usage of the ethanolic solution (96% (*w*/*w*)) as a solvent which is classified as sterilization agent [43]. All the composites fabrication stages were conducted using conditions and facilities designed to prevent microbial contamination. The sterility of the composites was verified via membrane filtration test [44]. No growth of bacteria confirmed the sterility of the composites.

#### 2.7.2. Antimicrobial Activity

Modified agar diffusion test [45] was performed to verify the ciprofloxacin potency against *Staphylococcus aureus* (ATCC 6538) released from the composites (EC/CIP, EC/PDMS/CIP, EC/MCM-41-CIP, EC/PDMS/MCM-41-CIP) over time. All composites were examined in triplicates. To confirm *Staphylococcus aureus* susceptibility to CIP, MIC, and minimal bactericidal concentration (MBC) were determined via serial dilution method [46]. The composites were placed onto the surface of Petri dishes (one composite per dish), then completely covered with a liquefied Mueller-Hinton (MH) agar (45 °C), and pre-incubated for 1 h at 37 °C. Then, a thin layers of liquefied MH agar (45 °C) inoculated with a suspension of *S. aureus* at density of 10^6^ CFU/mL were poured onto the surface of MH agar plates and incubated for another 24 h at 37 °C. Bacterial growth zones of inhibition (ZOI) were measured and photographed, then the composites were transferred aseptically into the new Petri dish and the whole procedure was repeated. After each 24 h of incubation the composites were transferred to the new Petri dish and covered with a freshly prepared MH agar inoculated with *S. aureus*, then incubated according to the description above. The procedure was repeated each day until the ZOI disappeared. The areas of ZOI were calculated using the image processing program (Image J).

#### 2.7.3. Cytotoxicity Assay

Human fetal osteoblastic cell line was cultured in 1:1 mixture of Ham’s F12 Medium Dulbecco’s Modified Eagle’s Medium (DMEM/F12), with 2.5 mM L-glutamine (without phenol red), 15 mM HEPES, and sodium bicarbonate, supplemented with 10% fetal bovine serum and penicillin/streptomycin (100 U/mL/100 μg/mL) at 34 °C in a humidified atmosphere of 5% CO_2_. Medium was replaced every 2–3 days. Cells were passaged for a maximum of 3–4 months post resuscitation and regularly tested for mycoplasma contamination by the two methods: DNA staining with 4,6-diamidino-2-phenylindoledihydrochloride (DAPI) and MycoAlertTM Mycoplasma Detection Kit. Cells were transduced for stable expression of enhanced GFP (eGFP) with pLVTHM plasmid (Addgene, Watertown, MA, USA). 

The in vitro cytotoxicity of the composites (EC/CIP, EC/PDMS/CIP, EC/MCM-41-CIP, EC/PDMS/MCM-41-CIP) was examined by direct contact test, according to the ISO Standard 10993-5 [47], which is focused on the physical interaction between examined materials and the cell monolayer. However, in case of drug delivery systems, the results may be also related to the drug released into the cell culture medium. Therefore, CIP-free analogues such as EC#, EC/PDMS# and EC/PDMS/MCM-41# and various concentration of CIP aqueous solutions (10, 20, 40, 80, 160 μg/mL) were also examined for comparative purposes. The frequently used proliferation tests based on addition of dyes (such as MTT or WST-1) may lead to false negative results due to unspecific interaction of the dye with tested material (e.g. dye adsorption onto the material surface) [48]. Thus, we decided to perform eGFP fluorescence-based assay. Since the eGFP fluorescence is lost after the cell death, the cells viability may be evaluated as fluorescence intensity [49]. 

According to the ISO Standard 10993-5 the specimen of analyzed material should cover one tenth of cell layer surface, thus the composites were cut into the circle-shaped samples with the diameter of 3.5 mm. For the test, cells were seeded in 48-well plate at a concentration of 3 × 10^4^ per well. After 24 h of incubation (34 °C, 5% CO_2_), the medium from each well was removed, specimens of composites were carefully placed onto the cell layer, 500 µL of fresh medium was added into each well and incubated for 72 h (34 °C, 5% CO_2_). Cells cultured without any specimen were used as a control. Images were obtained with Axiovert 200 fluorescent microscope equipped with AxioCam MRm digital camera (Zeiss, Oberkochen, Germany). Before the quantitative determination of cells viability, specimens were removed from each well and the conditioned medium was replaced with the fresh one. Fluorescence intensity was measured with excitation/emission at 485/528 nm using Synergy H1 microplate reader (BioTek, Winooski, VT, USA). Evaluation of the influence of CIP solutions on osteoblasts viability was performed in the same manner using the increasing concentration of drug instead of the specimens of composites. 

Data was presented as the mean ± standard deviation for three independent experiments. Statistical analysis was performed by Student’s *t*-test using STATISTICA 13.3 software (Statsoft, Kraków, Poland). The results were considered to be statistically significant when *p* value was <0.05 vs. control.

## 3. Results and Discussion 

### 3.1. The Synthesis of MCM-41 and CIP Adsorption onto Its Surface

MCM-41 in the powder form was successfully synthesized by using structure directing sol-gel method [50]. The mean amount of CIP adsorbed onto each 1 g of MCM-41 was 131 ± 5 mg that corresponded to 65 ± 2.5% adsorption efficiency. The adsorption of positively charged CIP onto negatively charged surface of MCM-41 [51] resulted in the change of zeta potential value from −12.02 ± 0.91 mV to 11.04 ± 0.04 mV for MCM-41 and MCM-41-CIP, respectively. The change of the surface zeta potential after drug adsorption onto the mesoporous silica is well-known phenomenon [52,53]. The negative charge of MCM-41 was caused by the dissociation of residual surface silanols (≡Si–OH ⇄ SiO^−^ + H^+^) in HCl solution (pH = 3.5) which acted as the adsorption sites for positively charged CIP molecules. 

The changes in the mesoporous structure of MCM-41 and MCM-41 after CIP adsorption were also confirmed by the nitrogen adsorption-desorption data (Appendix A). The decrease in the surface area, pore diameter, and pore volume of MCM-41 material by the factors of 1.22, 1.19 and 1.30, respectively, was observed. This phenomenon resulted from the pore blocking by the adsorbed drug molecules.

### 3.2. Fabrication of Composites

The composites were succesfully fabricated via solvent-evaporation blending method. All of the composites were characterized by the diameter of 11 ± 1 mm. For the composites containing of 0.79 ± 0.05 mg freeze-dried CIP (Table 1: EC/CIP and EC/PDMS/CIP) the thickness was in the range of 40–45 µm, whereas, for the composites containing MCM-41-CIP in amount of 6.0 ± 0.02 mg corresponding to 0.79 of CIP (Table 1: EC/MCM-41-CIP and EC/PDMS/MCM-41-CIP), the thickness increased up to 50–55 µm. The observed increase of thickness composite was related to the addition of MCM-41. 

### 3.3. Composites Physicochemical Properties

#### 3.3.1. Molecular Structure

The molecular structure of composites was investigated by FTIR technique. Figure 1 shows the spectra of precursors (EC, PDMS, freeze-dried CIP, MCM-41, MCM-41-CIP) as well as the spectra of the final composites: EC/CIP, EC/PDMS/CIP, EC/MCM-41-CIP, EC/PDMS/MCM-41-CIP. For the MCM-41 typical vibrations modes of Si–O–Si at 1097 cm^−1^ and 802 cm^−1^, Si–OH at 969 cm^−1^, and Si–O at 463 cm^−1^ were observed [54]. In comparison to the MCM-41 spectrum, the spectrum of MCM-41-CIP presented additional bands at 1715 cm^−1^, 1494 cm^−1^, and 1459 cm^−1^, which were also observed in the spectrum of freeze-dried CIP confirming the drug adsorption onto the MCM-41. These three bands were respectively attributed to the stretching vibrations of C=O, C–H, and aromatic C=C of CIP [55]. 

The well-defined peak of CIP at 1722 cm^−1^ was observed in the spectra of both EC/CIP and EC/PDMS/CIP composites (Figure 1). These spectra also presented bands characteristic for EC (2972 cm^−1^ (C–H), 1375 cm^−1^ (C–H), and 1052 cm^−1^ (C–O–C) for EC/CIP composite, and 2964 cm^−1^ (C–H), 1377 cm^−1^ (C–H) for EC/PDMS/CIP) [56]. The presence of PDMS in EC/PDMS/CIP composite was confirmed by the characteristic peaks at 1261 cm^−1^ and 802 cm^−1^ corresponding to the stretching vibrations of Si-C, and 1024 cm^−1^ attributed to stretching vibration of Si–O [57]. The spectra of composite containing adsorbed CIP onto MCM-41: EC/MCM-41-CIP and EC/PDMS/MCM-41-CIP showed the bands specific for all the components presented in the composites, excluding bands characteristic for CIP. It was explained by the CIP amount below the *detection limit* of FTIR method. However, the identification of CIP in MCM-41-CIP after the adsorption process (MCM-41-CIP spectrum, Figure 1) confirmed the CIP loading in the more complex composites. 

#### 3.3.2. Morphological and Structural Analysis

To characterize morphology and structural composition of obtained composites the SEM/EDX and XRD analyses were performed (Figure 2). To better visualize the alteration of morphology and crystallinity before and after polymer coating, the SEM/EDX images and XRD patterns of uncoated freeze-dried CIP and MCM-41-CIP were also presented. The morphological structure of freeze-dried CIP was highly heterogenous with observed irregularities in the crystals size and shape, whereas the MCM-41-CIP presented more homogeneous size distribution of spherical-shaped particles (the detailed particle size analysis of MCM-41-CIP was presented in Appendix A). After coating, the SEM images of composites containing freeze-dried CIP (EC/CIP, EC/PDMS/CIP) revealed elongated crystals with the sharp edges of CIP distributed in polymer blends (Figure 2c,e). However, the composites containing MCM-41-CIP (EC/MCM-41-CIP and EC/PDMS/MCM-41-CIP) showed the formation of compact and continuous polymer phase with homogeneously distributed MCM-41-CIP particles (Figure 2d,f). The significant differences in surface morphology between the composites with PDMS (EC/PDMS/CIP and EC/PDMS/MCM-41-CIP) and without PDMS (EC/CIP and EC/MCM-41-CIP) were observed (Figure 2, insets I-IV). As previously reported [40], based on the optical profilometry results, the surface roughness of these composites increases as function of PDMS. Therefore, this phenomenon explained the differences in SEM surface results. 

The EDX spectrum of freeze-dried CIP (Figure 2a) presented the relatively high peak of fluorine (F) - element characteristic for this molecule. After coating the freeze-dried CIP using EC the peak derived from F was still observed confirming the presence of drug in the EC/CIP composites (Figure 2c). After addition of PDMS (EC/PDMS/CIP, Figure 2e) the EDX profile showed additional peak of silicon (Si) element. Thus, the EDX spectra of the composites containing freeze-dried CIP confirmed the presence of particular components (EC, CIP, and PDMS). In the EDX spectra of composites containing MCM-41-CIP (Figure 2d,f) peaks of Si and O may be also attributed to MCM-41 (Figure 2b). There was no peak corresponding to CIP observed in these spectra which may be explained by fact that drug molecules were loaded into the mesopores that were additionally coated with the EC/PDMS blend. 

To examine the crystallinity of the composites, the XRD patterns of freeze-dried CIP and MCM-41-CIP before and after polymer coating were compared (Figure 2g). The XRD pattern of freeze-dried CIP showed highly crystalline structure with characteristic, more intense peaks at 8°, 15°, 21° and 27° 2*θ*. The XRD pattern of MCM-41-CIP presented the broad halo in the range of 15–35° derived from the amorphous silica [58] with two well-defined peaks at 8 and 27° 2*θ* characteristic for the CIP, suggesting a semi-crystalline structure of CIP adsorbed onto MCM-41. On the other hand, after MCM-41-CIP coating with EC or EC/PDMS blend, the EC/MCM-41-CIP and EC/PDMS/MCM-41-CIP composites were characterized by two diffraction haloes in the 2*θ* range of 10–15° and 15–30° derived from the amorphous polymer blends and silica, respectively, with no diffraction peaks derived from CIP. It suggested that the MCM-41-CIP was successfully coated with amorphous polymer blend. After the coating of CIP with EC or EC/PDMS blend the significant reduction in CIP crystallinity was observed revealing semi-crystalline nature of CIP in the EC/CIP and EC/PDMS/CIP composites, most probably as a consequence of the changes in overall background scattering of X-rays due to the presence of amorphous polymer coating blend. It should be noted that there were no significant differences in XRD patterns between composites with PDMS and PDMS-free composites.

#### 3.3.3. Texture Analysis

The texture properties were examined for all the final composites (EC/CIP, EC/PDMS/CIP, EC/MCM-41-CIP, EC/PDMS/MCM-41-CIP). For comparative purpose, the analogues without CIP and MCM-41-CIP, such as EC and EC/PDMS, were also tested. The values of rupture force, elasticity, firmness, weight, and thickness were presented in Table 2. The results were expressed as the means of 6 measurements with standard deviations. Rupture force referred to the force required to produce a major break of a sample, whereas the elasticity defined the distance at break. Based on these parameters, the values of firmness were calculated. The EC/CIP was characterized by relatively low rupture force (0.48 ± 0.16 N) and elasticity (0.58 ± 0.12 mm) which resulted in the low firmness (0.68 ± 0.22 N mm^−1^). Addition of PDMS (EC/PDMS/CIP) resulted in the increase of firmness to 0.97 ± 0.41 N mm^−1^. The same phenomenon was observed for the composites containing MCM-41-CIP. Thus, EC/MCM-41-CIP was characterized by the value of 2.5 ± 0.76 N mm^−1^, whereas for the composites with PDMS (EC/PDMS/MCM-41-CIP) the significant increase was observed (3.22 ± 0.81 N mm^−1^). The results of texture study were in good accordance with the values of weight and thickness. The firmness increased with the increase of the weight and thickness. 

#### 3.3.4. Thermal Analysis

To determine the effect of polymer coating on thermal stability, the TG curves and DSC profiles of uncoated freeze-dried CIP and MCM-41-CIP, and final composites (EC/CIP, EC/PDMS/CIP, EC/MCM-41-CIP, EC/PDMS/MCM-41-CIP) were compared (Figure 3). MCM-41 was also examined for comparative purpose. The thermal decomposition of ciprofloxacin (Figure 3a) was characterized by two mass losses, first in the temperature range of 25–250 °C (23% of its mass) and, the second, between 250–550 °C (51%). The detailed description of this process can be found in the literature [59]. On the other hand, the total weight loss of MCM-41-CIP was 17% what confirmed the calculated amount of drug adsorbed onto MCM-41. Pure MCM-41 was characterized by the high thermal stability with small mass loss (3%) below 100 °C which may resulted from the evaporation of water absorbed in the silica channels [60]. 

All final composites were stable in the temperature range of 25–330 °C (Figure 3b). Mass loss of EC/CIP was observed above 330 °C, and was equal to 98%, whereas in case of EC/MCM-41-CIP decrease to 69%. Addition of PDMS to the composites slightly changed their thermal stability. Heating the EC/PDMS/CIP and EC/PDMS/MCM-41-CIP above 330 °C resulted in mass loss of 99% and 72%, respectively. 

The DSC profiles (Figure 3c,d) were in accordance with the observations described above. The profile of freeze-dried CIP (Figure 3c) presented the endothermic peak ca. 90 °C which probably resulted from the evaporation of water, whereas, the endothermal decomposition started above 150 °C. This decomposition was not observed in the curve of MCM-41-CIP indicating the increase of drug thermal stability after its adsorption onto MCM-41. 

The DSC profiles presented in Figure 3d confirmed the high thermal stability of the composites. There was no peak attributed to CIP decomposition noticed up to 250 °C. Above this temperature small phase transition changes were observed, especially for the composites without MCM-41, suggesting the beginning of the decomposition process. It is not seen for EC/PDMS/MCM-41/CIP, thus this composite seemed to be the most stable. The difference in the stability temperature obtained from TG and DSC (330 °C and 250 °C, respectively) results from the differences in the measurement conditions. 

#### 3.3.5. In Vitro Ciprofloxacin Release

The CIP release profiles of obtained composites and MCM-41-CIP before coating were presented in Figure 4. Both the EC/CIP and EC/PDMS/CIP composites were characterized by the high burst release (89.0 ± 2.5% and 85.4 ± 0.8%, respectively) of CIP after first 24 h of release study. Comparatively, the 80 ± 3.2% of adsorbed CIP was released from MCM-41-CIP during the first 24 h (Figure 4b). These three samples (MCM-41-CIP, EC/CIP and EC/PDMS/CIP) were characterized by complete drug release after 6–7 days with no significant differences between them. The use of CIP adsorbed onto MCM-41 in EC/MCM-41-CIP composite instead of freeze-dried CIP resulted in reduction of burst release by the factor of 2.2 from 89.0 ± 2.5% (for EC/CIP) to 38.8 ± 3.1% (for EC/MCM-41-CIP) after first 24 h (Figure 4a). Moreover, the addition of PDMS into the polymer blend in EC/PDMS/MCM-41-CIP resulted in higher reduction of CIP burst release by the factor of 4.4 compared to EC/MCM-41-CIP (from 38.8 ± 3.1% to 8.8 ± 1.2%) (Figure 4a). As described in our previous studies [38,40], the PDMS addition into the composites resulted in a prolonged drug release due to the occlusion of mesopores of drug-loaded silica particles by hydrophobic PDMS chains. The PDMS-occluded silica particles impeded the penetration of dissolution medium into the composites, and hence the further dissolution and release of drug loaded into the pores were slowed down.

As previously reported [40], based on Higuchi and Korsmeyer–Peppas models, the CIP release from proposed non-disintegrating ethylcellulose-based composites was diffusion-controlled and followed zero-order kinetics after the burst stage. Consequently, the composites obtained in this study were characterized by both high R^2^ for Higuchi model (from 0.936 to 0.993) and by Korsmeyer–Peppas release exponent n in the range of 0.17 to 0.54 proving a simple diffusion or quasi-diffusion-controlled drug release mechanism (Appendix A).

The kinetic parameters of the zero-order release for all obtained composites were also calculated using Equation (3) and presented in Table 3.(3)$$Q=Q_{0}+k_{0}t\;$$
*Q* [%]—the fraction released by time t [days], *Q*_0_ [%]—the initial fraction of released drug in burst stage, *k*_0_—zero-order release constant [% of dose released per day].

The release data confirmed that the prolonged release of CIP with negligible burst was presumably associated with the drug entrapment into the pores of MCM-41 which were additionally coated by EC/PDMS blend [40]. Thus, the presence of PDMS and MCM-41-CIP seemed to be a crucial factor for ensuring the zero-order release with highly reduced burst stage what was further correlated in both the microbiological and cytotoxicity assays. 

### 3.4. Composites Biological Evaluation

#### 3.4.1. Bacterial Growth Inhibition Assay

The modified agar diffusion test was performed for all obtained composites. To verify antimicrobial activity of antibiotic released from the composites the areas of ZOI were determined. The susceptibility of *Staphylococcus aureus* (ATCC 6538) to CIP was expressed by MIC and MBC values which were 0.125 µg/mL and 0.25 µg/mL, respectively. In Figure 5, the average areas of ZOI with standard deviations and corresponding representative images were presented. The areas of resulting ZOI corresponded to the amounts of antibiotic diffused into the MH agar and indicated that the value of MIC was reached. After first 24 h of incubation the composites containing freeze-dried CIP, such as EC/CIP and EC/PDMS/CIP, were characterized by the ZOI area of 48.5 ± 8.42 cm^2^ and 37.8 ± 7.23 cm^2^, respectively. Similar area of ZOI (36.2 ± 6.8 cm^2^) was observed for the PDMS-free composites containing MCM-41-CIP (EC/MCM-41-CIP). The addition of PDMS to the composites containing MCM-41-CIP reduced ZOI to 12.66 ± 2.59 cm^2^ which was in good accordance with the significantly reduced initial burst in the in vitro release results (Figure 4). 

The EC/CIP and EC/PDMS/CIP composites lost the antimicrobial activity against *S. aureus* after day 6 and 7, respectively. It was considered to be not sufficient duration of antimicrobial activity in context of the treatment of *osteomyelitis* using bone antibiotic drug delivery systems. By contrast, the ZOI determined for the composites containing MCM-41-CIP occurred after each cycle of incubation until day 20 (EC/MCM-41-CIP) and day 30 (EC/PDMS/MC-41-CIP). The results of bacterial growth inhibition were in good accordance with the in vitro release results (Figure 4). In both studies, the EC/PDMS/MCM-41-CIP was recognized as the most promising candidate for local bone antibiotic delivery system characterized by release the smallest amount of CIP in the burst stage and significantly sustained release. The occurrence of ZOI after each cycle of incubation confirmed that the amount of CIP released from the composites sufficiently exceeded MIC, thus, EC/PDMS/MCM-41-CIP remained active against *S. aureus* for 30 days. The discrepancy between zero-order release kinetics (Table 3, 21% of cumulative drug amount released after 30 days) and the loss of ZOI after 30 days of incubation may be explained by high viscosity of MH agar medium that might have clogged both the nanopores in ethylcellulose structure and the CIP-loaded mesopores in MCM-41 particles which inhibits the further drug release to MH agar.

#### 3.4.2. Cytotoxicity Assay

The quantitative results and representative images of cytotoxicity assay against human fetal osteoblasts of all obtained composites with drug (EC/CIP, EC/PDMS/CIP, EC/MCM-41-CIP, EC/PDMS/MCM-41-CIP) were presented in Figure 6. The tested circle-shaped specimens which covered 10% of cell layer surface contained approx. 60 µg of CIP. Cytotoxicity of various concentration of referenced CIP aqueous solutions and the CIP-free analogues such as EC#, EC/PDMS#, and EC/PDMS/MCM-41# was also shown for comparative purposes. 

The CIP cytotoxicity assay revealed the concentration-dependent effect (Figure 6b), with the calculated IC_50_ at concentration of 79 µg/mL. The statistically significant reduction in cell viability to 75 ± 7.4% and 86 ± 6.3% (*p* < 0.05) was observed for the MCM-41-free composites containing freeze-dried CIP: EC/CIP and EC/PDMS/CIP, respectively (Figure 6a). It was well correlated with the in vitro release results which revealed the highest burst release of CIP for EC/CIP and EC/PDMS/CIP composites (approx. 89.0% and 85.0%, respectively, Figure 4). There was no significant cytotoxic effect observed for the composites containing MCM-41-CIP (Figure 6a, EC/MCM-41-CIP and EC/PDMS/MCM-41-CIP) which were characterized by significantly reduced initial burst release (Figure 4) suggesting that the amounts of CIP released into the culture medium were non-toxic to osteoblast. The CIP-free analogues (EC#, EC/PDMS#, and EC/PDMS/MCM-41#) did not impede the osteoblasts growth as well (Figure 6a). Therefore, the negative impact on osteoblasts viability observed for EC/CIP and EC/PDMS/CIP was possibly related to the initial release of higher dose of drug into the medium (compared to the others composites) what also correlated with drug release profiles (Figure 4).

## 4. Conclusions

Herein, the composites composed of freeze-dried CIP or MCM-41-CIP coated with EC or EC/PDMS blend were successfully obtained via solvent-evaporation blending method. The physicochemical properties, drug release profiles, antibacterial activity and cytocompatibility with osteoblasts of the prepared composites were assessed to select the most promising candidate for further in vivo evaluation as implantable self-contained antibiotic delivery system or functional coating for implant surfaces. 

The most beneficial properties were reported for the formulation composed of ciprofloxacin adsorbed onto MCM-41 coated with blend of ethylcellulose and polydimethylsiloxane (EC/PDMS/MCM-41-CIP). It presented significantly higher thermal stability than the composites containing freeze-dried CIP (instead of MCM-41-CIP) and composites not containing PDMS. Moreover, EC/PDMS/MCM-41-CIP was characterized by sustained release rate with the lowest initial burst of ciprofloxacin. The microbiological and biological evaluation was in correlation to the in vitro release results, revealing the 30-day maintenance of antimicrobial activity and excellent cytocompatibility with human fetal osteoblasts.

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
