# Peer review of "Silica-Polymer Composites as the Novel Antibiotic Delivery Systems for Bone Tissue Infection"

_pharmaceutics, 2019, doi:10.3390/pharmaceutics12010028_

Round 1
Reviewer 1 Report
In this manuscript, authors have studied the development of functional nanocomposites for local bone antibiotic delivery systems. This article describes how to develop ciprofloxacin-loaded nanocomposites (mesoporous silica nanocarriers/ethylcellulose/polydimethylsiloxane) by using a solvent-evaporation blending strategy against Staphylococcus aureus. In my view, this manuscript (ID: pharmaceutics-668671) can be accepted after considering the following comments:
1. Authors should annotate FTIR and XRD peaks in Figure 1 and 2. The figures need to be corrected by showing the bonds (i.e., instead of 1024 cm-1: ν Si-O and other peaks) and crystalline structures.
2. Although authors mentioned ciprofloxacin release from nanocomposites should be a diffusion and zero order kinetic, the ciprofloxacin release kinetic from all nanocomposites should be checked by other models (i.e., Higuchi and Korsmeyer-Peppas) in order to confirm it (i.e., R2 bigger than 99).
3. According to Figure 5, the burst release from the composites (EC/CIP and EC/PDMS/CIP) delivered a therapeutic mount of ciprofloxacin and efficiently stopped the bacterial growth, and the designed bacterial growth inhibition assay has trivialised the important outcome of this study. Authors should eliminate the burst release effect and repeat the assay after the burst stage in order to highlight the importance of the sustained release from the EC/MCM-41-CIP and EC/PDMS/MCM-41-CIP. In this case, both EC/CIP and EC/PDMS/CIP composites should not be able to provide a long-term bacterial growth inhibition compared to the developed nanocomposites.
Author Response
With co-authors we would like to thank the Reviewer for the thoughtful comments. Detailed answers to comments are found in attached file. Please note that our comments below are outlined in blue text, changes made directly into the manuscript are in red.

Reviewer 2 Report
Authors designed a long-acting ciprofloxacin formulation consisting of mesoporous silica microparticles and ethylcellulose-based polymer for local bone tissue infection. Authors well characterized the final silica-polymer composites containing drug in vitro and tested its antibacterial growth. The whole idea of this study is very interesting and valuable to the research in this field. The manuscript is also well written and presented with promising results, however, there are some limitations (lack of necessary characterizations, missing of a critical control group and inadequate justification of some results, etc.) that prevent the current manuscript from acceptance. Authors have to address those concerns carefully to improve the quality of this manuscript. The detailed comments are listed below for authors’ reference.
The characterization of MCM-41-CIP should be supplemented, such as size, distribution and morphology. The blanks in figure2a-2f should be explained. The XRP patterns for both EC/MCM-41-CIP and EC/CIP is missing in figure2g-2h. A critical reference group (MCM-41-CIP) should be included in many tests, including in vitro drug release and bacteria growth inhibition, to justify the protective effect of EC-based polymer on burst release of MCM-41-CIP. The detailed information for Ethyl cellulose, such as viscosity and molecular weight, should be defined and provided. There are some other characters for final EC/MCM-41-CIP, such as hardness and plasticity, which need considering and testing. Those characters will be valuable for the clinical application. Because of the application of cell study and antibacterial assay, the sterilization of the final products should be noted and explained. The chosen ratio here of EC/PDMS in the polymer matrix should be justified and discussed. The EC concentration in ethanol (5%) should be justified. Is there any consideration for high concentration of EC? The photos of final products are better to be included in this manuscript. The lower drug release from EC/PDMS/MCM-41-CIP compared with EC/MCM-41-CIP should be explained with additional discussion, which justify the function of PDMS in this MCM-polymer matrix. The function of PDMS should be discussed. “After first 24 h of incubation, the biggest areas of ZOI were observed for EC/CIP, EC/PDMS/CIP, and EC/MCM-41-CIP (~36.2 – 48.5 cm2) what was correlated with huge bursts in in vitro release study (Figure 4)”. The description is not clear here. Authors should revise that. Additionally, the Data should be presented as the mean ± standard deviation here. Non-treated cell should be included as negative control in figure 6.Author Response
With co-authors we would like to thank the Reviewer for the thoughtful comments. Detailed answers to comments are found in attached file. Please note that our comments below are outlined in blue text, changes made directly into the manuscript are in red.

Round 2
Reviewer 1 Report
-No revision is needed.
Reviewer 2 Report
I am fine with authors' replies.
No more revision is needed from my side.